# Prenatal Low Testosterone Levels Induced by *DNAH8* Dysfunction Leads to Urethral Fusion and Male Differentiation Abnormalities

**DOI:** 10.3390/biomedicines13123032

**Published:** 2025-12-10

**Authors:** Zhiwei Peng, Yao Li, Yaping Wang, Mingming Yu, Yiqing Lyu, Fang Chen, Yichen Huang, Yu Ding

**Affiliations:** Department of Urology, Shanghai Children’s Hospital, School of Medicine, Shanghai Jiao Tong University, Shanghai 200062, China; pengzhiweiys@163.com (Z.P.); liyaor@163.com (Y.L.); wyp_0730@163.com (Y.W.); ymmnjmu@hotmail.com (M.Y.); tristan_ren@hotmail.com (Y.L.); chenfang01@sjtu.edu.com (F.C.)

**Keywords:** hypospadias, single-cell sequencing, testis, prenatal testosterone levels, external genitalia, development

## Abstract

**Background:** Hypospadias development is influenced by prenatal androgen levels, with genetic factors typically playing a significant role. Through whole-exome sequencing, we found that rare damaging variants in *DNAH8* (dynein axonemal heavy chain 8) were significantly enriched in hypospadias cases. However, the role of *DNAH8* deficiency in hypospadias pathogenesis remains unclear. **Objectives:** This study aimed to clarify the function of *DNAH8* in urethral development and fusion. **Materials and Methods:** Using CRISPR/Cas9, we generated *DNAH8* knockout mice and employed a multi-disciplinary approach to evaluate urogenital development, male differentiation, testosterone levels, steroid biosynthesis gene expression, and cellular changes in fetal testes and external genitalia. **Results:**
*DNAH8* knockout mice presented abnormal masculinization phenotype, and fetal mice exhibited urethral fusion defects and hypoplastic glans during early urethral development. *DNAH8* knockout was found to reduce prenatal testosterone levels and steroid biosynthesis in the testes. Based on single-cell sequencing and multicolor immunofluorescence, we demonstrated that in the early stage of fetal testis development, the loss of *DNAH8* function affected the differentiation of Sertoli and steroidogenic cell lineages, thereby impairing testosterone synthesis ability during the masculinization programming window. Meanwhile, we identified two key distal glans cell populations that cause abnormal urethral fusion and hypoplastic glans. Furthermore, *DNAH8* knockout could synergistically interact with low-dose endocrine-disrupting chemicals, increasing the incidence of urethral fusion defects at E16.5, and led to clear hypospadias phenotypes at E18.5. **Conclusions:** Loss of *DNAH8* delays differentiation of Sertoli and steroidogenic lineages, reduces prenatal testosterone, and, with environmental exposure, increases hypospadias risk.

## 1. Introduction

Hypospadias is one of the most common congenital urogenital malformations in boys. Although the reported prevalence shows large differences, in general, the reporting trend in recent decades has been increasing in all countries, reaching 80 per 10,000 male births in some parts of Europe and the United States [1,2]. The cause of hypospadias is still unclear [3], but the main and concentrated view is that the key to the development of hypospadias is androgen deficiency, especially testosterone (T) [4], which is genetically caused by genetic and environmental factors during the development of fetal sexual organs [5,6]. When testosterone levels decrease during the human masculine programming window (MPW; 8–14 weeks of gestation), masculinization often appears abnormal [7]. Therefore, hypospadias in children is often accompanied by shortened anogenital distance (AGD) [8], incomplete testicular descent [9], micropenis [10], increased digit ratio (2D:4D) [11] and other masculinizing phenotype abnormalities.

Previous genetic studies have shown that approximately 60% of hypospadias cases are genetically related, and other patients are also genetically susceptible [6,12]. Some recent progress has been made in the use of genome sequencing technology. For example, Mads Melbye et al. [13] used GWAS to report hypospadias risk-associated variants in a population of European ancestry. Kalfa et al. [14] strategically constructed a next-generation sequencing panel containing the most comprehensive 365 genes to date. However, hypospadias has a high degree of genetic heterogeneity, and previous studies have shown that, unlike other types of genetic variation, rare deleterious mutations have more adverse effects on protein function and are more likely to cause disease [15], highlighting the need for further genetic exploration.

To overcome the above problems, we previously performed whole-exome sequencing (WES) and RNA sequencing on a large hypospadias cohort and found a group of cilia-associated outer dynein arm heavy chain (ODNAH) genes with rare deleterious mutations in 24% of hypospadias patients, enabling us to identify a novel risk gene for hypospadias—*DNAH8* [16]. Here, we established a *DNAH8* knockout mouse model and verified the effect of the variation on the development of hypospadias and abnormal masculinization. Furthermore, we analyzed the cells of the testis and external genitalia during the key embryonic stage of urethra formation and masculinization via multiomics methods such as protein mass spectrometry and single-cell RNA sequencing (scRNA-seq). Based on these data, we developed a single-cell atlas and described its communication network, obtaining evidence for the regulatory effect of *DNAH8* on the differentiation process of steroidogenic cells. Our results provide convincing evidence that *DNAH8* is a risk gene for hypospadias, characterize its function, and offer new strategies for the prevention and treatment of hypospadias.

## 2. Materials and Methods

### 2.1. DNAH8 Knockout Mouse Model

Homozygote whole-body *DNAH8* knockout mice (C57BL/6J) were established by Shanghai Southern Model Biotechnology, China. CRISPR/Cas9 technology was employed, utilizing non-homologous recombination repair to introduce mutations, leading to a frameshift in the DNAH8 protein coding sequence and ultimately resulting in a loss of function. One sgRNA targeting intron 3 of the *DNAH8* gene was 5′-AGACTTAAGACCTCCGAGG-3′, and the other sgRNA targeting intron 4 of the *DNAH8* gene was 5′-CATTACTTAUGUACACTCTGGG-3′. Each female mouse was mated with another male adult mouse at 5 pm and the morning when the vaginal plug was found was considered embryonic day 0 (E0). The mice were housed in polycarbonate cages in a controlled environment with a 12 h light/12 h dark cycle (lights at 7:00 am/7:00 pm), with free access to drinking water and food. All experimental procedures were approved by the Animal Ethics Committee of Shanghai Jiao Tong University Affiliated Sixth People’s Hospital (Shanghai, China; Approval No. DWSYHZ2020-002).

### 2.2. Morphological Measurements

Fetal mice were obtained at E16.5, the gonads and external genitalia were examined through a Leica M165FC stereomicroscope (Leica Microsystems, Wetzlar, Germany), and male fetal mice were selected. The limbs and tail were dissected to obtain an unobstructed view of the anogenital region, and the gross morphology was observed under a Leica DM6 microscope (Leica Microsystems, Wetzlar, Germany). Anogenital distance (AGD) was quantified by measuring the length from the caudal base of the genital tubercle to the anterior part of the anus. Meanwhile, to better measure the digit ratio and the descending position of the testes, adult mice (7 weeks old) were selected. We used method from previous studies to measure digit ratio, and the 2D and 4D lengths of each claw were measured from the midpoint of the toe root to the tip of the toe, excluding nails [11]. The kidney and testis were freed through surgical dissection, and the descending position of the testis was compared by measuring the distance from the upper pole of the testis to the lower pole of the kidney with a caliper. The above values were measured by three surveyors, and the averages were taken.

### 2.3. ELISA Testosterone Measurement

The testes of E16.5 and E18.5 male fetal mice were obtained through surgical dissection, and the samples were ground via a KZ-III-F high-speed and low-temperature tissue grinder (Servicebio, Wuhan, China) to obtain testis homogenates. Testicular testosterone concentrations were measured through testosterone enzyme-linked immunosorbent assay (ELISA) kits (Beyotime, Shanghai, China) according to the manufacturer’s instructions. The measurement of testosterone levels was repeated three times.

### 2.4. Immunofluorescence Staining

The testes of male fetuses at the designated experimental time points were harvested through surgical dissection, fixed in 4% paraformaldehyde overnight, dehydrated and embedded. Then sections with a thickness of 6 mm were cut out using a paraffin microtome (RM2235; Leica Biosystems, Nussloch, Germany). Paraffin sections were stained with appropriate antibodies (Key resources table, Antibodies) via tyramide signal amplification (TSA) according to the manufacturer’s instructions for the multicolor immunofluorescence scanning (GP2035) kit (Servicebio, Wuhan, China). The nuclei were counterstained with 40,6-diamidino-2-phenylindole (DAPI). The sections were scanned under a Zeiss Axioscan 7 digital scanner (Zeiss, Oberkochen, Germany) for panoramic imaging, and the images were observed and shot via Case Viewer software (V2.4, 3DHISTECH Ltd., Budapest, Hungary).

### 2.5. Label-Free Quantitative Mass Spectrometry-Based Proteomics and LC-MS/MS Analysis

Male fetuses were harvested from E16.5 pregnant mice, and bilateral testis tissues were obtained by dissection. For each WT and homozygous (HOM) sample, three technical replicates were analyzed. Liquid chromatography–tandem mass spectrometry (LC–MS/MS) analysis, protein identification and quantification were performed by Applied Protein Technology (APL, Shanghai, China). The raw MS data of each sample were merged and searched by MaxQuant 1.6.14 software (Max Planck Institute of Biochemistry, Munich, Germany), and the screening criteria for credible proteins and peptides were both set at a False Discovery Rate (FDR) ≤ 0.01. The identification and quantitative analysis of the differentially expressed proteins were performed via the DAVID database.

### 2.6. RT-qPCR

Testis tissues from E16.5 and E18.5 male fetal mice were ground using a KZ-III-F high-speed and low-temperature tissue grinder (Servicebio, Wuhan, China), and total RNA was extracted via a RNeasy Mini Kit (QIAGEN, Dusseldorf, Germany). Complementary DNA (cDNA) synthesis was performed using a Prime Script™ RT Kit (Takara, Kyoto, Japan). Quantitative PCR were performed via TB Green^®^ Premix Ex Taq™ II (Takara, Kyoto, Japan) on a QS7 Fast Real-Time PCR System (Thermo Fisher Scientific, Waltham, MA, USA) according to the manufacturer’s instructions. The primer pairs used are listed in Appendix A.

### 2.7. Single-Cell RNA-Seq

Single-cell processing was performed using a Chromium Controller (10× Genomics, Pleasanton, CA, USA) according to the manufacturer’s instructions. The procedures for single-cell RNA-seq library preparation, sequencing, and subsequent data analysis are described in detail in the Appendix A.

### 2.8. Quantification and Statistical Analysis

Statistical analysis was performed via SPSS statistical software version 21.0 (SPSS Inc., Chicago, IL, USA) and Prism 9 (Prism 9 by GraphPad Software, Boston, MA, USA). All bars represent the mean ± SD. The Shapiro–Wilk test was used to assess normality, and Levene’s test was performed to evaluate homogeneity of variances. Differences among groups were evaluated by Student’s *t* test, the Mann–Whitney U test, Chi-square test or the variance test for continuous variables. Differences were considered to be significant at *p* < 0.05 and are represented by *, those at *p* < 0.01 are indicated by **, and those at *p* < 0.001 are represented by ***. Statistical details are described in the legend of each experiment.

## 3. Results

### 3.1. Generation of the DNAH8 KO Mouse Model

We used CRISPR/Cas9 technology to completely remove exon 4 to establish a *DNAH8* knockout mouse model (Figure 1A), and the genotypes of the mice were identified via PCR (Figure 1B). Male mice in the HOM group were mated for more than 2 months and failed to give birth to offspring, confirming that biallelic *DNAH8* mutation causes sterility [17]. Immunofluorescence staining revealed that DNAH8 in the testes of HOM mice was absent compared with wildtype (WT) mice (Figure 1C).

### 3.2. Loss of DNAH8 Function Leads to Abnormalities in the Development of the Male Urethra and the Masculinization Phenotype

In the early differentiation stage of urethral penis development, we observed abnormal urethral development in the heterozygous (HET) group (18.9%, n = 7/37) and hypospadias in the HOM group (25%, n = 5/20) at E16.5 (Figure 1D). No morphological abnormalities of the external genitalia were observed in the WT group (n = 0/35). Moreover, compared with those in WT mice, micropenis was more likely to be observed in the HET and HOM groups (27% and 45%, respectively), and this difference was caused mainly by the shortness of the glans penis (Figure 1E). Interestingly, the urethral dysplasia phenotype observed in HOM groups in the early differentiation stage (E16.5) was not observed in the late differentiation stage (E18.5, n = 25) and later (Appendix A).

Compared with WT, we observed that HET and HOM groups had smaller AGD (WT 613.72 ± 100.87 vs. HET 522.03 ± 77.17 vs. HOM 491.39 ± 63.57 μm, *p* = 0.002) (Figure 1F) and larger digit ratio (WT 0.96 ± 0.01 vs. HET 0.98 ± 0.02 vs. HOM 1.00 ± 0.02, *p* < 0.001) (Figure 1G), especially in HOM. Only 2 out of 20 WT male mice had abnormal testicular descent (10%); however, in HOM male mice, the proportion of abnormal testicular descent increased sharply (Figure 1H), reaching 41% (7/17). The descending position of the tests showed significant differences (WT 1.52 ± 0.16 vs. HOM 1.31 ± 0.29 cm, *p* = 0.027). Our observations indicated that the deletion of *DNAH8* could cause abnormalities in the early differentiation stage of urethral development and the masculinization phenotype.

### 3.3. Overview of the General Cell Populations in the Fetal External Genitalia

We generated a single-cell transcriptome atlas of external genitalia from E16.5 male fetal mice with an abnormal urethral developmental phenotype (Figure 2A). After quality control and preprocessing, 23,577 cells were finally obtained (WT = 11,605, HOM = 11,972) from external genitalia samples (Appendix A). After principal component analysis (PCA) and Uniform Manifold Approximation and Projection (UMAP) dimensionality reduction clustering (resolution k = 0.5), 26 cell clusters were obtained (Figure 2B and Appendix A). And the expression of specific marker genes in each cell cluster is shown in Appendix A. Through the gene expression profiles of known cell type markers of the external genitalia [18,19] (Appendix A) and the annotation of cell types by marker genes determined in the PanglaoDB and Enrichr databases, we identified 10 major cell populations of the external genitalia (Figure 2C). These include the corpus cavernosum cells (CC), distal dorsal glans cells (DDG), distal ventral glans cells (DVGs), endothelial cells (End), epithelial cells (Epi), macrophages cells (MAC), prepuce cells (PP), preputial gland cells (PG), and smooth muscle cells (SMCs). The expression of specific marker genes in each cell subgroup is shown in Figure 2E and Appendix A.

### 3.4. Distal Glans Cell Significant Deficiency Is a Major Pathological Feature in the Early Differentiation Stages of DNAH8 KO Fetal External Genitalia

Mesenchymal cells accounted for the greatest proportion of the external genitalia at E16.5. Moreover, the heterogeneity of external genitalia cells in the WT and HOM groups was mainly concentrated in the two groups of cell populations at the distal glans penis. Compared with the WT group, the DDG and DVG were significantly reduced in the HOM group (Figure 2B–D). In mice, the urethra at the distal glans is formed through fusion events [20,21,22]. It can be seen that the DDG and DVG at the distal glans play key roles in the occurrence of urethral formation.

Next, we calculated the differential gene expression of the DDG and DVG in the WT and HOM groups and found that these two cell groups presented no functional changes (Figure 2F and Appendix A). Further enrichment analysis (Figure 2G) found that the pathways enriched in the cell populations in the distal glans included Hedgehog, TGF-β, angiogenesis, epithelial–mesenchymal transition, Wnt-β, and Notch. These familiar pathways are all responsible for penis development and the promotion of urethral closure [23]. Moreover, these two cell populations at the distal glans express AR and are regulated by androgen levels. Since DNAH8 is expressed in the testis but not in the external genitalia, we speculated that the loss of *DNAH8* function affects the function of the testis, thereby reducing the quantity of DDG and DVG and causing the abnormal phenotype we observed above.

### 3.5. DNAH8 Deletion Decreases Prenatal Testosterone Level and Steroid Biosynthesis

We performed label-free quantitative mass spectrometry on the testes of WT and HOM mice at E16.5 (Appendix A). A total of 480 differentially expressed proteins (fold change > 2 and *p* < 0.05) were detected (Figure 3A,B). GO biological process analysis of the differentially expressed proteins (Figure 3C) revealed that the biological process (BP) terms were related to cellular processes, biological regulation, and metabolic processes. KEGG pathway and pathway enrichment analyses revealed that the differentially enriched pathways included the FoxO signaling pathway, steroid hormone biosynthesis pathway, and Hedgehog signaling pathway (Figure 3D), which are closely related to the level of testosterone secretion by testicular Leydig cells and the data also showed significant downregulation of steroid biosynthesis genes (Figure 3E).

Compared with those in WT fetal mice, the testosterone levels in the testes of HET and HOM fetal mice were significantly lower at early differentiation stage (E16.5, WT 1417.60 ± 228.60 vs. HET 994.53 ± 174.82 vs. HOM 586.27 ± 210.10 pg/gonad, *p* = 0.008) (Figure 4B). Interestingly, this decrease in testosterone levels was alleviated in the late differentiation stage (E18.5, WT 1336.60 ± 189.41 vs. HET 1112.17 ± 169.13 vs. HOM 933.63 ± 200.17 pg/gonad, *p* = 0.098) (Figure 4C). As a target of androgen action, the transcription level of AR also showed the same change (Appendix A). We further detected the expression of genes associated with steroid synthesis (Figure 4A). At E16.5, the absence of *DNAH8* led to significant decreases in the Hedgehog signaling pathways in the upstream pathway of steroidogenesis, as well as the transcription levels of *Star*, *Cyp11a1* and *Cyp17a1* in the steroidogenesis pathway in fetal Leydig cells. Surprisingly, this phenomenon was reversed at E18.5. We speculated that the absence of *DNAH8* downregulated the transcription levels of key genes in the Hedgehog and steroidogenic signaling pathway in the early stage of urethral development. While in the late stage of E18.5, this abnormal decrease was reversed because of the compensatory mechanism.

### 3.6. Characterization of the Cellular Composition of the Fetal Testis

We isolated cells from the testis of E16.5 male fetal mice with abnormal urethral developmental phenotype (Figure 5A). After quality control and preprocessing, 24,359 cells were finally obtained (WT = 12,546, HOM = 11,813) from testis samples (Appendix A). After PCA and UMAP dimensionality reduction clustering, 26 cell clusters were obtained (Appendix A).

Through the annotation of cell types on the basis of the gene expression profiles of known cell type markers [24,25,26,27], we identified 15 major cell populations of the testis (Figure 5C and Appendix A). Early-interstitial progenitors (e-IPs), endothelial cells (End), fetal Leydig cells (FLCs), germ cells (GCs), immune cells (IM), late-interstitial progenitors (l-IP), macrophages (MACs), Mesonephric mesenchyme cells 1 (MM1), Mesonephric mesenchyme cells 2 (MM2), Mesonephric tubes (MTs), neutrophil cells (NCs), pre-supporting cells (p-SCs), perivascular cells (PV), Sertoli cells (SCs), and surface epithelium cells (SE) are included, and the expression of specific marker genes of each cell subgroup is shown in Figure 5D.

Pre-supporting cells (17%) further differentiated into SCs at E11.5, and their relative density ranged from 17% at E12.5 to 24% at E16.5. Mesenchymal progenitor cells persisted throughout the fetal period, ranging from 39% at E12.5 to 58% at E16.5 of the total number of testis cells. These cells were the origin of FLCs and adult Leydig cells (ALCs) and strongly and specifically expressed the Wnt5a [24,25]. As expected, our data revealed that the proportion of mesogenic progenitor cells in the testis at E16.5 was the highest at approximately 50% (e-IP and l-IP cells), followed by SCs at approximately 20%. The heterogeneity of fetal mice testis cells in the WT and HOM groups in the UMAP maps was mainly concentrated in Mesonephric Mesenchymal cells (Figure 5B,C,E). Compared with the WT, MM1 was significantly reduced, and MM2 was significantly increased in HOM group.

### 3.7. Loss of DNAH8 Delays the Differentiation of Cells in the Sertoli and Steroidogenic Lineages

The Sertoli cell lineage and the steroidogenic cell lineage originate from a common heterogeneous progenitor cell pool derived from the coelom epithelium (CE) and mesenchymal progenitor cells of mesonephros origin. [24,25,28,29,30] Our data indicated that the majority of heterogeneous progenitor cell pool this time were mesonephric mesenchymal cells, which could be further divided into two subclusters on the basis of specific markers, including Wnt5a+/Arx−/Postn+/Rspo3−(MM1) and Wnt5a+/Arx−/Postn−/Rspo3+(MM2). Further pseudotime analysis (Figure 6A–F) also confirmed that MM1 was the root of the differentiation trajectory, which was converted to e-IP through MM2 and could differentiate into SCs, FLCs, and l-IPs (possibly representing the dormant progenitor cell population at the time of ALC origin). Moreover, compared with the WT group, the HOM group lacked the differentiation peak of MM1-MM2 but had the peak of MM2 differentiation. We further performed interaction analysis of intercellular communication via Cell Chat. Notably, the MM clusters (MM1 and MM2) showed strong signal export ability (Figure 6G,H), indicating that they have important regulatory functions in the testicular microenvironment and are also the source of differentiated progenitor cells, while PV cells and SCs show typical signal input patterns. Among these signaling interactions, many canonical regulatory pathways have been identified (Appendix A). With the loss of DNAH8 function, MM1 cells are rapidly consumed, whereas MM2 cells increase strongly and its output signals increase significantly. On the basis of the above evidence, we propose a reasonable hypothesis: loss of DNAH8 function is associated with abnormalities in the MM1–MM2–e-IP differentiation pathway. For further verification, we took E14.5 and E16.5 as two key time points and observed the changes in each cell population in time through the population-specific markers selected by scRNA-seq. Multicolor immunofluorescence imaging revealed that, compared with the WT group, the HOM group exhibited a significant increase in MM1 cells in the testes at E14.5, followed by a sharp decrease at E16.5, with a corresponding increase in MM2 cells (Figure 6I). Additionally, at E14.5, the IP in the testes of the HOM group was significantly lower than in the WT group; however, this trend dramatically reversed at E16.5 (Figure 6J). At 14.5, the number of SCs and FLCs that are ultimately responsible for the secretion of testosterone in HOM group was significantly less than that in WT group. While at E16.5, with the changes in MM and IP mentioned above, the number of SCs and FLCs in the HOM group significantly increased, showing no significant difference compared to the WT group (Figure 6K and Appendix A).

Overall, these data indicate that the absence of DNAH8 could delay the downward differentiation of MM1 cells at early E14.5, resulting in a decrease in the number of SCs and FLCs, thereby resulting in the low testosterone level observed at E16.5. However, this effect was only temporary, the consumption of the MM1 progenitor cell pool accelerated at E16.5, and compensatory differentiation toward MM2—e-IP—SC/FLC was performed so that the testosterone levels could be compensated. This compensatory phenomenon may explain why the abnormal urethral developmental phenotype at E18.5 was not observed. This finding is also consistent with the study by Martin J. Cohn’s group [31], who reported that in *Gli3* knockout fetal mice, the observed hypospadias phenotype in early stage was restored with development.

### 3.8. The Synergistic Effect of DNAH8 Deficiency and Low-Dose DEHP Can Lead to Hypospadias

To model the interaction between genetic predisposition and environmental changes during pregnancy, we administered low-dose DEHP orally to pregnant mice. Our findings revealed that low-dose DEHP alone did not induce significant abnormalities in penile and urethral development. However, in *DNAH8* knockout mice, low-dose DEHP markedly increased the incidence of urethral fusion defects at E16.5 (Figure 7A) and led to clear hypospadias phenotypes at E18.5 (Figure 7B,C).

## 4. Discussion

The pathogenesis of hypospadias is complex, and recent research suggests that the high heterogeneity of hypospadias (i.e., variability in clinical presentation and genetic background) is mainly due to the interaction between risk genes and pathogenic genes or risk genes and environmental factors. However, there is still a lack of stable and reliable hypospadias susceptibility animal model to further elucidate the above hypothesis. Moreover, at single-cell resolution, our understanding of how risk genes affect cell heterogeneity and function in the testis and external genitalia is still limited. In this study, we constructed a novel hypospadias mouse model and used scRNA-seq to create single-cell transcriptome profiles of testis and external genitalia tissues, which revealed the tissue cellular composition, biological characteristics and regulatory signaling network, and to better understand the mechanism of risk genes.

In our previous study, a novel risk gene for hypospadias, *DNAH8*, was discovered through large-scale WES [16]. In recent years, some groups have conducted studies on the biological function of *DNAH8*. Zhang F et al. [17] reported for the first time that *DNAH8* is associated with multiple morphological abnormalities of the flagella (MMAF), and subsequent studies also revealed that the function deletion mutation of *DNAH8* can lead to MMAF and cause male infertility [32,33,34]. Evidently, *DNAH8* is closely related to testis function and, combined with the above WES results, prompted us to explore its role in the development of the penis and urethra. Here, we constructed *DNAH8* knockout (*DNAH8* KO) mice via CRISPR/Cas9 technology.

We verified the sterility of HOM male mice and found that *DNAH8* KO mice at the late stage of differentiation at E18.5 and after birth presented no abnormalities in the external genitalia or urethra, which was almost the same as that of the control mice. There were no observed phenotypes of hypospadias or reduced penile development, which was consistent with previous studies showing that male mice heterozygous or homozygous for the *DNAH8*, *DNAH9*, and *DNAH17* pathogenic mutations did not have the hypospadias phenotype [17,33,35,36]. However, when we advanced the observation time to the early differentiation stage at E16.5, the HOM male fetal mice presented hypospadias and micropenis phenotypes, and abnormalities were also observed in the further masculinization phenotype.

The development of external genitalia is a highly coordinated process involving the participation of multiple cellular populations, which contribute to tissue formation and ultimately result in the development of a functional organ. The three primary tissue types involved include the corpus cavernosum, glans, and prepuce. So, we used scRNA-seq to explore which cell changes led to the observed abnormal phenotype. In an elegant recent study, Amato et al. [19] created a single-cell atlas of the development of fetal mouse external genitalia and divided the glans into three groups of cells according to their unique markers: distal dorsal cell (*Inhba*, *Alx4*, *Syt13* and *Dkk1*), distal ventral cell (*Dlx5*, *Scube1*, *Notum*, *Hand2*, and *Ahr*) and proximal glans cell (*Gli1*, *Foxl2*, *Foxf2*, and *Foxd1*). Our data indicated that the loss of *DNAH8* function ultimately significantly reduced the cell proportion in the distal glans and that the tubular urethra in mice was remodeled mainly by fusion at the distal glans [20,21,22]. Different from the proximal spongy body, the glans tissue is heterogeneous and experiences significant morphological and transcriptional changes during development, and this can explain the abnormal urethral fusion and micropenis phenotype we observed in E16.5 fetal mice: a significantly reduced DDG population caused short glans, whereas the DVG caused abnormalities in ventral urethral fusion events. In addition, we identified several signaling pathways in these two cell groups that may play important roles in the process of glans growth and urethral fusion: Hedgehog, TGF-β, angiogenesis, epithelial–mesenchymal transition (EMT), Wnt-β and Notch. Hedgehog and Wnt-β signaling have been shown to play key roles in the early stage of penis development. The Hedgehog signaling pathway is considered a possible key factor in the initial development of external genitalia and sexual dimorphism in coordination with androgen signaling [4,37,38]. Wnt signaling is located downstream of Hedgehog signaling, while β-catenin activity is detected in the initial stage of the development of the cloacal membranous tubercle [39] and is critical for the regulation of the growth of external genitalia and the masculinization of mesenchymal material near the urethra [40]. EMT is an important process in urethral embryogenesis, the fusion of the fetal mouse urethra extends from the proximal to the distal and the epithelial cells in the urethral suture are remodeled into mesenchymal cells through EMT [23]. TGF-β can activate Smad proteins through phosphorylation to regulate the development of EMT and affect the formation of the tubular urethra [41,42]. Baskin’s latest study revealed that CD31-positive blood vessels reach the distal of the glans at 8 weeks in the human fetus and are critical for promoting the subsequent closure of the male urethra [43]. Notch signaling plays a role in the developmental programs of most organs and tissues and can act on endothelial cells, pericytes and vascular smooth muscle cells (VSMCs) in early fetal development to play an important role in angiogenesis and its stability. [44,45] Our data complement the knowledge and understanding of the key cells and their functional pathways in the ventral urethral fusion of fetal mice.

Androgens during the fetal period are mainly synthesized and secreted by the testes. The appearance of FLCs starts at E12.5 and the development of the urethra and penis as well as the correct masculinization process are extremely dependent on their ability to secrete testosterone [24,25]. The urethral plate begins to tubularize, forming the urethral groove, and with the production of testosterone, the lateral edges of the urethral groove (urethral folds) fuse at the ventral midline to form a tubular urethra [20]. The masculinization process is caused mainly by androgens in the masculinization programming window (E12.5–16.5) [46]. Meanwhile, DNAH8 is not expressed in the external genitalia, but is highly expressed in the testis. Therefore, we wondered whether the abnormal phenotype observed at E16.5 was caused by insufficient testosterone levels during this period.

To answer this question, we used protein mass spectrometry to detect differences in the biological function of the testes of fetal mice at E16.5. These results were consistent with our expectations: steroid biosynthesis was significantly decreased, which was also confirmed by the measurement of intratesticular testosterone levels via enzyme-based hormone (ELISA). Our evidence revealed that the hedgehog pathway and the steroidogenic pathway were damaged in the testes of the HOM fetal mice at E16.5, whereas the transcription levels were restored at E18.5. Since previous studies have shown that the hedgehog pathway is located downstream of the androgen receptor [47,48], we further observed AR transcription levels at E16.5 and E18.5 and observed the same trend. Moreover, in fetal mouse testes, AR was expressed mainly in SCs and FLCs [49]. These results support the hypothesis that the absence of DNAH8 affects the ability of SCs and FLCs to synthesize steroids in the early differentiation stage, thereby downregulating testosterone levels.

To clarify this hypothesis, we used scRNA-seq to generate a single-cell transcriptome map of E16.5 testis tissue. Our results indicated that at E16.5, a common heterogeneous progenitor pool that differentiated into the Sertoli cell lineage and the steroidogenic cell lineage still exists, which is composed mainly of mesonephric-derived mesenchymal progenitors (MM1 and MM2), and the MM cluster is the main signal output source in the testis microenvironment. Furthermore, at E16.5, we found that instead of SCs and FLCs, MM cells were the most different cell population between the WT and HOM groups. Compared with that in WT fetal mice, the progenitor cell pool of MM1 cells was sharply consumed in the HOM group, which accelerated differentiation, increased the number of MM2 cells and significantly increased their output signals. Therefore, we selected earlier E14.5 and E16.5 testis to further observe the cell populations via multicolor immunofluorescence. The results demonstrated that the absence of *DNAH8* could delay the downward differentiation of MM1 in the early masculinization window, resulting in a decrease in the number of SCs and FLCs and the observed low levels of *AR*, hedgehog, and steroid transcripts and low testosterone levels at E16.5 were not surprising. However, this effect is only temporary. At E16.5, the MM1 progenitor cell pool had already been consumed at an accelerated rate to compensate for downward differentiation to increase the number of SCs and FLCs. Therefore, this downregulation was reversed at E18.5. The reason for this result may lie in the function of the cilium gene to which *DNAH8* belongs. Cilia are tiny hair-like structures on the cell surface and can be divided into two types: motile cilia and primary cilia. Primary cilia are widely present in almost all mammalian cells, especially in developing embryos. Consequently, previous studies on testis development mainly focused on primary cilia. Elanor N Wainwright et al. reported that it can determine the size of the urogenital ridge and divide into several testicular cords [50]. Marie Berg Nygaard et al. demonstrated that primary cilia mediate signals through paracrine factors (such as Dhh), participate in the regulation of the recruitment or differentiation of testis Leydig cells, and play a key role in development and function of somatic cells and Leydig cells in the testis [51]. *DNAH8* is a component of the axonal dynein complex in motile cilia, as shown in the Appendix A, it is present mainly in the flagella of fully developed sperm in testis. However, we found the expression of DNAH8 in the MM and IP cell populations, although the expression level was not very high (Appendix A). This is not surprising because, during certain developmental stages, some pluripotent stem cells and primitive progenitor cells may form motile cilia. These cilia usually appear within specific developmental windows and disappear after completing their specific functions and this phenomenon is particularly obvious during the development of the nerves, respiratory tract and reproductive system [52,53]. This temporary characteristic is highly consistent with the temporary delay in steroid cell differentiation we observed in the testes of HOM fetal mice. Moreover, there is coordination between motile cilia and primary cilia in the processes of cell signaling and development [54]. Our studies showed that the loss of *DNAH8* function delayed the differentiation of the progenitor cell pool to the Sertoli cell lineage and the steroidogenic cell lineage by regulating cilia function during a short prenatal window of motile cilia emergence, which ultimately led to the decline of testosterone levels during the prenatal masculinization programming window (E12.5–16.5). However, the harm caused by this inheritance is limited, with the end of the window period for the appearance of motile cilia, its own compensatory effect can rectify the previous abnormalities and restore testosterone levels. Consequently, under the subsequent influence of normalized testosterone levels, no abnormal phenotypes were observed in fetal mice at E18.5.

A review of previous studies on hypospadias revealed that many animal studies in which genes were knocked out did not reveal the hypospadias phenotype in male offspring. Therefore, Baskin et al. proposed the following hypothesis: a single gene mutation that causes urethral penis malformation also has a fetal lethality. In general, the gene mutations associated with the development of urethral penis malformation affect only the genetic susceptibility of the fetus, and the generation of the phenotype requires a second hit of other intrauterine factors [55]. The combination of genetic factors and other intrauterine risk factors may explain the pathogenesis of hypospadias better [5,6]. Endocrine-disrupting chemicals (EDCs), characterized by their persistence in the environment and strong bioaccumulation potential, have been increasingly recognized for their detrimental effects on the development of the urogenital system [56]. Di-2-ethylhexyl phthalate (DEHP) is a common endocrine-disrupting chemical. Animal studies have demonstrated that prenatal exposure to DEHP can result in abnormal male sexual differentiation in neonate mice, even hypospadias [57]. A study by Xiang Zhou and colleagues further corroborated these findings, showing that DEHP may interact with genetic factors in the etiology of hypospadias [58]. However, the DEHP doses used in those studies [42,59] (500/750/1000 mg/kg/day) were much higher than those observed in everyday life and clinical settings [60,61]. Therefore, we chose a lower dose that is more realistic, and the results showed that the synergistic effect of *DNAH8* deficiency and low-dose DEHP can lead to hypospadias, and the *DNAH8* KO mouse is an ideal genetic susceptibility model of hypospadias. Future studies will focus on the interaction of genetic factors with EDCs.

## 5. Conclusions

In the early stage of fetal testis development, the loss of *DNAH8* function delayed the differentiation of the Sertoli cell lineage and steroidogenic cell lineage, resulting in a reduction in the number of SCs and FLCs and thus a decrease in testosterone synthesis and secretion. Ultimately, by affecting androgen-dependent DDG and DVG cells, abnormal urethral fusion and small glans were induced (Appendix A). This study emphasized that *DNAH8* is a risk gene of hypospadias, presented supplementing evidence of the function of cilia genes and provided a stable and reliable hypospadias susceptibility animal model for future studies on the interaction between genetic factors and EDCs. Further research such as generating patient-specific point mutation knock-in mice models is needed to strengthen our findings and determine whether human *DNAH8* variants recapitulate hypospadias phenotypes.

## Figures and Tables

**Figure 1 biomedicines-13-03032-f001:**
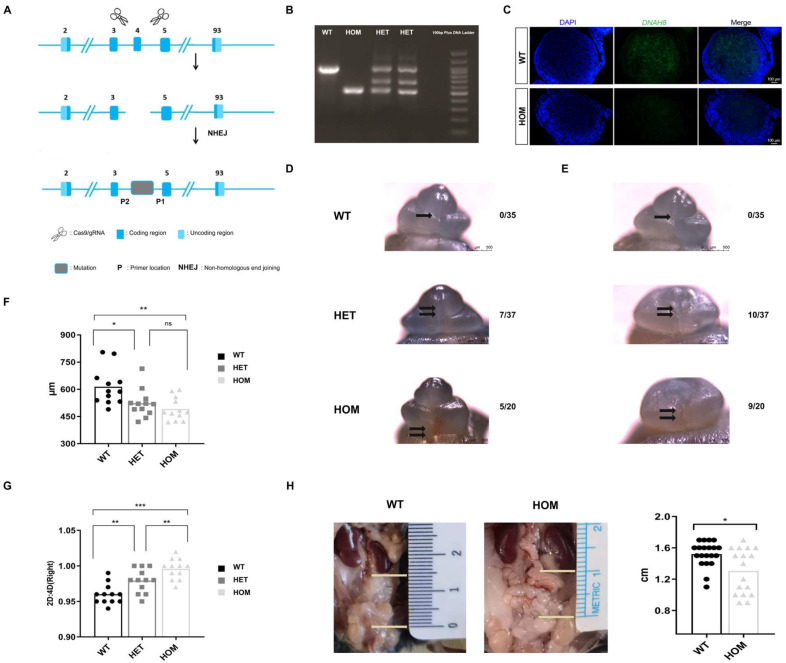
Construction of *DNAH8* KO mice, abnormality of urethral development and masculinization phenotype. (**A**) Schematic diagram of constructing *DNAH8* knockout mice using CRISPR/Cas9. (**B**) PCR genotyping of *DNAH8* wildtype (WT, *DNAH8*^+/+^) (938 bp), heterozygous (HET, *DNAH8*^+/−^) (three bands), and homozygous (HOM, *DNAH8*^−/−^) (499 bp) mice. (**C**) Immunofluorescence of DNAH8 (green) in testes obtained from WT and HOM with E16.5 mice. Nuclei are stained with DAPI (blue). Scale bar: 100 μm. (**D**,**E**) External genitalia in male embryos imaged in whole-mount using light microscopy at E16.5. No morphological abnormalities of the external genitalia were observed in the WT group, with a clear opening of the urethra (single black arrow), and the urethral plate on both sides had fused at the midline to form a closed urethra. No obvious urethral gap was found in the HET group and the urethral plates on both sides above the urethral opening did not fuse at the midline to form a closed urethral suture (double black arrowhead). HOM group showed significantly abnormal opening of the urethra (double black arrows), and similar to the HET group, the urethral epithelium on both sides did not fuse at the midline to form a closed urethral suture. (**F**) AGD in each group in the mouse model (n = 12 for each group, A one-way analysis of variance (ANOVA). (**G**) 2D:4D ratio of right hindlimb in each group in the mouse model (n = 12 for each group, ANOVA). (**H**) The descending position of the testes (the distance between the lower pole of the kidney and the upper pole of the testis) in male mice (20 mice in the WT group and 17 mice in the HOM group, Mann–Whitney U test). * *p* < 0.05, ** *p* < 0.01, and *** *p* < 0.001. ns: no significance. Created in BioRender. Ding, Y. (2025) https://BioRender.com/b36t765 Accessed on 5 December 2025.

**Figure 2 biomedicines-13-03032-f002:**
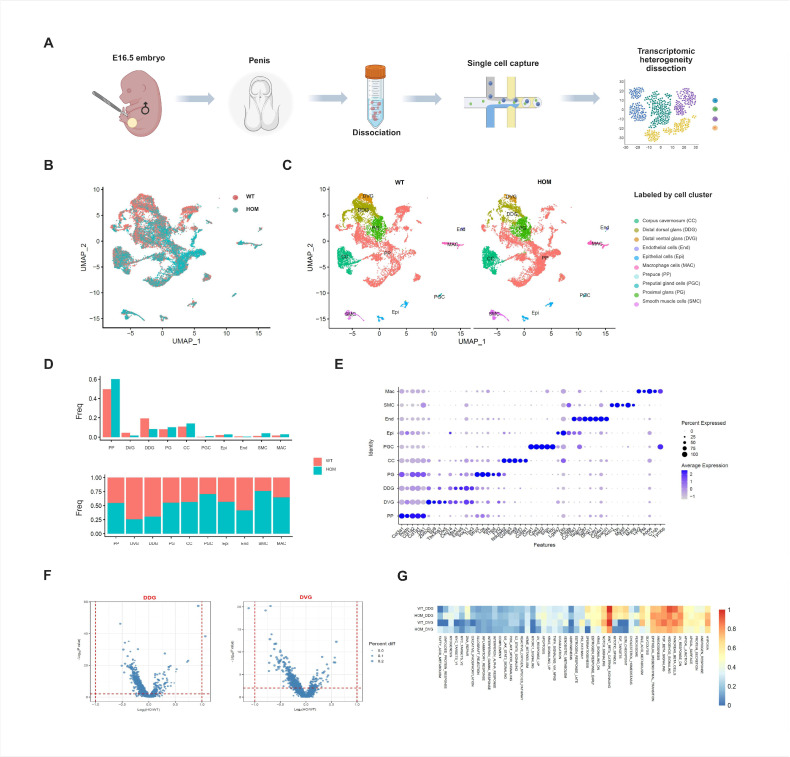
Single-cell sequencing analysis of external genitalia. (**A**) Schematic illustration of the experimental workflow in this study. (**B**) Uniform Manifold Approximation and Projection (UMAP) plots of all external genitalia cells from E16.5 WT and HOM mice (red = WT, blue = HOM). (**C**) All external genitalia cells from E16.5 WT and HOM mice are colored according to their types (10 cell clusters). (**D**) Bar plot showing the proportion of each subcluster in WT and HOM group. (**E**) Dot plot showing the expression levels of genes used for annotation in each cluster. (**F**) Volcano plot of differentially expressed genes (DEGs) in WT and HOM group. (**G**) Pathway enrichment analysis of DDG and DVG cell populations in the external genitalia of WT and HOM groups. Created in BioRender. Ding, Y. (2025) https://BioRender.com/62qwao0 Accessed on 4 December 2025.

**Figure 3 biomedicines-13-03032-f003:**
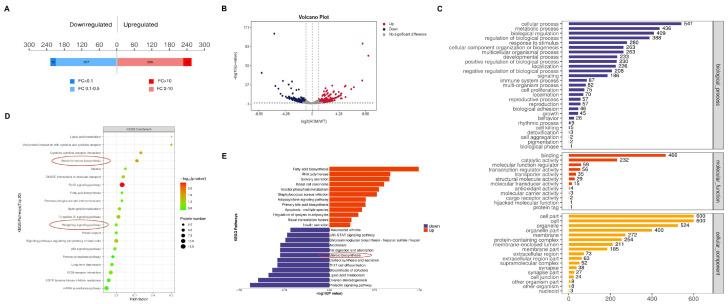
Proteomics analysis of the mice testes. (**A**) A total of 480 differentially expressed proteins were detected (fold change > 2 and *p* < 0.05) in the testes of HOM mice compared with WT mice, of which 254 were upregulated and 226 were downregulated. (**B**) Volcano plot of differentially expressed proteins detected in the testes of between the WT and HOM groups. (**C**) GO analysis (biological process, cellular component and molecular function) according to the DEGs shown as a bar diagram. (**D**) KEGG analysis according to the DEGs shown as a bubble diagram. Red circles: important pathways. (**E**) Label-free quantitative mass spectrometry-based proteomics demonstrated that *DNAH8* KO mice significantly dysregulated the biological process of steroid biosynthesis. Created in BioRender. Ding, Y. (2025) https://BioRender.com/97q8g46 Accessed on 4 December 2025.

**Figure 4 biomedicines-13-03032-f004:**
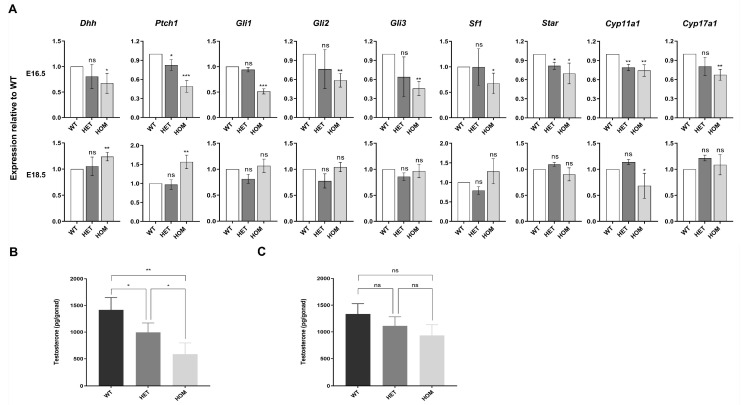
Hedgehog and steroidogenic pathway gene expression and testosterone levels in mice testes. (**A**) RT-qPCR comparing each group testis mRNA expression for hedgehog and steroidogenic pathway genes at E16.5 and E18.5 and expressed as fold change from wildtype control at each time point (Student’s *t* test). n = 3–4 biological replicates. (**B**) Intratesticular testosterone levels as measured by ELISA in WT, HET, and HOM mice at E16.5. Three rounds of ELISA measurements, with each round including one pair of testes from WT, HET, and HOM fetal mice derived from different rounds of pregnant mice (ANOVA). (**C**) Intratesticular testosterone levels as measured by ELISA in WT, HET, and HOM mice at E18.5 (ANOVA). Three rounds of ELISA measurements, with each round including one pair of testes from WT, HET, and HOM fetal mice derived from different rounds of pregnant mice. Error bars are ± s.d. * *p* < 0.05, ** *p* < 0.01, and *** *p* < 0.001. ns: no significance. Created in BioRender. Ding, Y. (2025) https://BioRender.com/q43w858 Accessed on 4 December 2025.

**Figure 5 biomedicines-13-03032-f005:**
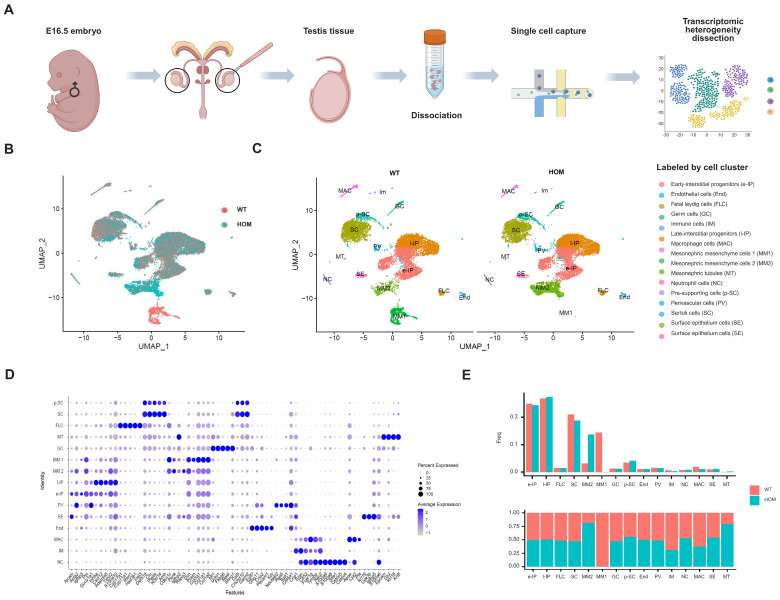
Characterization of mice testes by single-cell RNA sequencing. (**A**) Schematic illustration of the experimental workflow in this study. (**B**) Uniform Manifold Approximation and Projection (UMAP) plots of all testis cells from E16.5 WT and HOM mice (red = WT, blue = HOM). (**C**) All testis cells from E16.5 WT and HOM mice are colored according to their types (15 cell clusters). (**D**) Dot plot showing the expression levels of genes used for annotation in each cluster. (**E**) Bar plot showing the proportion of each subcluster in WT and HOM group. Created in BioRender. Ding, Y. (2025) https://BioRender.com/8ir7pef Accessed on 4 December 2025.

**Figure 6 biomedicines-13-03032-f006:**
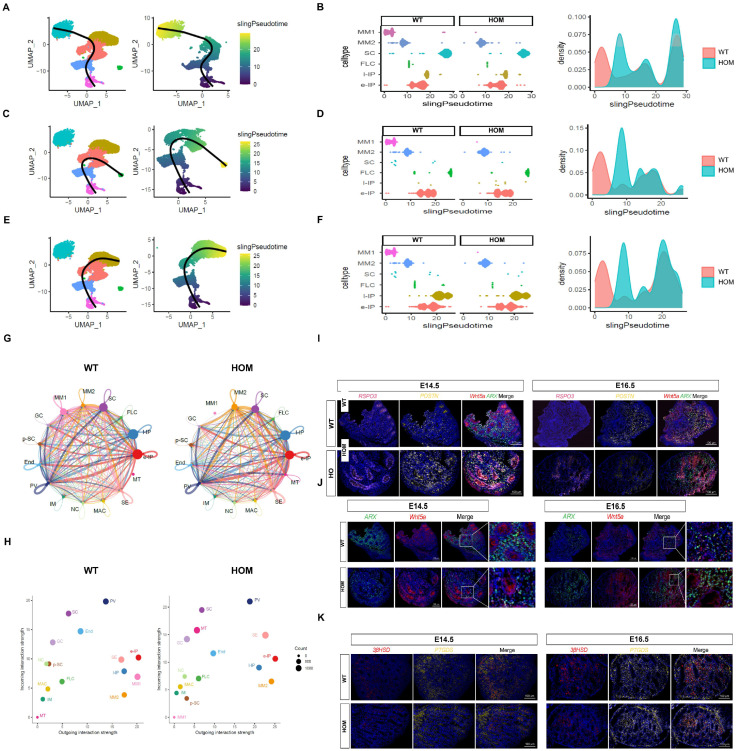
Loss of *DNAH8* delays the differentiation of cells in the Sertoli and steroidogenic lineages. (**A**–**F**) Pseudotime analysis confirmed that MM1 was the root of the differentiation trajectory, which was converted to e-IP through MM2 and could differentiate into SCs, FLCs, and l-IPs. compared with the WT group, the HOM group lacked the differentiation peak of MM1-MM2 but had the peak of MM2 differentiation. (**G**) Cell–cell communication signaling network among the fifteen clusters analyzed with CellChat in WT and HOM groups. The width of the lines indicates the number of pairs. Different colors represent different signal sources. (**H**) Cell clusters were located based on the count of their significant incoming (Y-axis) or outgoing (X-axis) signaling pattern. (**I**) Co-immunostaining for the MM1 cell marker Wnt5a+ (red)/Arx− (green)/Postn+ (yellow)/Rspo3− (pink) and the MM2 cells marker Wnt5a+ (red)/Arx− (green)/Postn− (yellow)/Rspo3+ (pink) in the testes of WT and HOM mice at E14.5 and E16.5. (**J**) Co-immunostaining for the interstitial progenitor cells marker Wnt5a (red) and Arx (green) in the testes of WT and HOM mice at E14.5 and E16.5. (**K**) Co-immunostaining for the FLC marker HSD3B (red) and the SC marker PTGDS (yellow) in the testes of WT and HOM mice at E14.5 and E16.5. Scale bar: 100 μm. Created in BioRender. Ding, Y. (2025) https://BioRender.com/2hreg7n Accessed on 4 December 2025.

**Figure 7 biomedicines-13-03032-f007:**
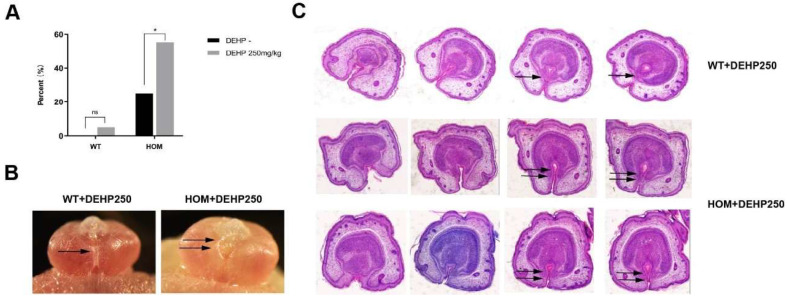
Endocrine-disrupting chemicals (EDCs) increase the incidence of hypospadias in *DNAH8* KO mice, and disrupt compensatory mechanisms. (**A**) The incidence of hypospadias from WT (n = 35), WT + DEHP 250 mg/kg (n = 20), HOM (n = 20) and HOM + DEHP 250 mg/kg (n = 29) group of E16.5 mice (Chi-square test). * *p* < 0.05, ns: no significance. (**B**) Gross anatomical images of the external genitalia in male embryos at E18.5. Single arrow marks the normal Urethral fusion; Double arrows indicate urethral plates on both sides above the urethral opening did not fuse at the midline. (**C**) H&E analysis of pathological changes in the external genitalia of E18.5 male mice (from the distal to proximal penis). In the WT + DEHP (250 mg/kg) group, as the penile tip transitions toward the base, the epithelial cells of the bilateral urethral plate and the mesenchymal cells migrate progressively toward the ventral midline (indicated by single arrows). This migration leads to the fusion of urethral folds and mesenchyme, forming a complete urethral lumen. In contrast, in the HOM + DEHP (250 mg/kg) group, the epithelial and mesenchymal cells of the bilateral urethral plate fail to fully fuse on the ventral side of the penis (indicated by double arrows), resulting in failed urethral tubulogenesis.

## Data Availability

All data generated or analyzed during this study are included in this published article. The raw sequencing data are available from the corresponding author upon reasonable request.

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
