# Peer review of "Prenatal Low Testosterone Levels Induced by DNAH8 Dysfunction Leads to Urethral Fusion and Male Differentiation Abnormalities"

_biomedicines, 2025, doi:10.3390/biomedicines13123032_

Round 1
Reviewer 1 Report
Comments and Suggestions for Authors
Comments to the Author: The article titled “Prenatal Low Testosterone Levels Induced by DNAH8 Dysfunction Leads to Urethral Fusion and Male Differentiation Abnormalities” is a well-conceived and technically robust study that significantly advances our understanding of the genetic and developmental mechanisms underlying hypospadias. Congratulations to the authors on conducting such a thoughtful and informative study.
However, there are several areas where the manuscript could be further strengthened:
1- The abstract is informative but could be tightened to emphasize the key mechanistic conclusion: loss of DNAH8 delays differentiation of Sertoli and steroidogenic lineages, reduces prenatal testosterone, and, with environmental exposure, increases hypospadias risk.
2- In abstract it is better to consider rewording “small glans” to “hypoplastic glans” for clinical precision
3- For All quantitative data (AGD, 2D:4D, testicular descent, urethral closure status) should include mean ± SD, exact P values, and the statistical test used. Consider showing effect sizes to complement P values.
4- Intratesticular testosterone levels are reported to be reduced at E16.5 and normalized at E18.5. Include the absolute values, sample sizes, and variance.
5- Consistent reporting of n-values and replicates for each figure and table should be consider. Include P values and the exact tests used in figure legends or a Methods subsection.
6- All figures should have high-resolution images, clear labels, and consistent color schemes. All supplementary figures/data should be referenced in the main text where relevant.
7- Please use “fetal Leydig cells (FLCs)” consistently (not “IPC” or “interstitial progenitors” interchangeably without definition)
Author Response
1- The abstract is informative but could be tightened to emphasize the key mechanistic conclusion: loss of DNAH8 delays differentiation of Sertoli and steroidogenic lineages, reduces prenatal testosterone, and, with environmental exposure, increases hypospadias risk.
Response: Thank you for your suggestion. We have revised the abstract as requested.
2- In abstract it is better to consider rewording “small glans” to “hypoplastic glans” for clinical precision
Response: Thank you for your suggestion. We have revised the abstract as requested.
3- For All quantitative data (AGD, 2D:4D, testicular descent, urethral closure status) should include mean ± SD, exact P values, and the statistical test used. Consider showing effect sizes to complement P values.
Response: Thank you for your suggestion. We have supplemented the results section and Figure legends based on your suggestions.
4- Intratesticular testosterone levels are reported to be reduced at E16.5 and normalized at E18.5. Include the absolute values, sample sizes, and variance.
Response: Thank you for your question. We have supplemented the results section and Figure legends based on your suggestions.
5- Consistent reporting of n-values and replicates for each figure and table should be consider. Include P values and the exact tests used in figure legends or a Methods subsection.
Response: Thank you for your suggestion. We have supplemented it in the Figure legends.
6- All figures should have high-resolution images, clear labels, and consistent color schemes. All supplementary figures/data should be referenced in the main text where relevant.
Response: Thank you for your suggestion. We have made relevant references in the main text.
7- Please use “fetal Leydig cells (FLCs)” consistently (not “IPC” or “interstitial progenitors” interchangeably without definition)
Response: Thank you for your question. We identified 15 major cell populations of the testis. Early-interstitial progenitors (e-IP), endothelial cells (End), fetal Leydig cells (FLC), germ cells (GC), immune cells (IM), late-interstitial progenitors (l-IP), macrophages (MAC), Mesonephric mesenchyme cells 1 (MM1), Mesonephric mesenchyme cells 2 (MM2), Mesonephric tubes (MT), neutrophil cells (NC), pre-supporting cells (p-SC), peri-vascular cells (PV), Sertoli cells (SC), and surface epithelium cells (SE) are included. Among them, fetal Leydig cells (FLCs) and interstitial progenitors (IP) are not the same type of cells. As stated in reference 23 (Ademi H, Djari C, Mayere C, Neirijnck Y, Sararols P et al. Deciphering the origins and fats of steroidal lineages in the mouse test. Cell Rep 2022; 39:110935.), IP is recognized as a precursor cell for steroidogenic cells. Early interstitial progenitors (e-IP) subsequently differentiate into fetal Leydig cells (FLCs), while late interstitial progenitors (l-IP) subsequently differentiate into adult Leydig cells (ALCs). According to standard specifications, the abbreviation for interstitial producers has been changed from IPC to IP.
Reviewer 2 Report
Comments and Suggestions for Authors
This manuscript investigates the functional role of DNAH8 in fetal urethral development, testosterone production, masculinization, and hypospadias susceptibility. The authors combine CRISPR mouse modeling, multi-omics, and histological work to build a mechanistic framework linking DNAH8 deficiency to transient prenatal hormonal disruption and urethral fusion anomalies. The manuscript is broadly suitable for journals focusing on developmental biology, urogenital disorders, pediatric urology, or reproductive genetics. For journals with very high mechanistic thresholds, some parts require deeper functional validation.
Abstract
abstract provides good overview of multi-layered approach. The phrasing “delayed differentiation of Sertoli and steroidogenic cell lineages” implies causality; however, causality is shown indirectly and may need softening (Lines 25–29).
Introduction
It is comprehensively literature coverage with strong justification for genetic exploration in hypospadias. The link between androgen deficiency and associated phenotypes is well stated but could benefit from referencing the timing of the masculinization programming window (L# 43-48).
The introduction to DNAH8 function relies heavily on infertility studies; more developmental context is needed.
Materials and methods
Methodological breadth is impressive and CRISPR design and genotyping steps are clearly stated. However, following points should be clarified in revision:
Proteomics section needs thresholding details for peptide identification.
single-cell processing lacks details on cell viability thresholds and batch correction approach.
Statistical analysis section is too generic; it requires clarity on exact tests for each figure.
Results
The description of urethral plate fusion is solid, but quantification of fusion abnormalities should be shown. The claim that abnormalities disappear by E18.5 may require more animals for statistical confidence.
Proteomic fold-change thresholds should be justified (Lines 247–256). The explanation for compensation is plausible but speculative; additional functional assays would help (Lines 264–275).
The MM1/MM2 progenitor axis is described well but some lineage intermediates (MM2 → e-IP → FLC) are inferred rather than experimentally validated.
DEHP Synergy Experiment: Dose justification is reasonable, but pharmacokinetics of DEHP in mice should be noted and sample sizes are relatively small. statistical tests used for incidence comparison should be stated. Also blinded scoring should be mentioned for histology description.
Discussion
This section Logically integrates findings. Following points should be addressed in revised version:
Clarify why phenotypes appear only at E16.5 but not E18.5 (Lines 426–431).
The argument about transient motile cilia requires citations showing motile cilia presence in early testis progenitors (Lines 514–529).
The gene×environment model discussion is excellent but could be condensed (Lines 538–562).
Author Response
Abstract
abstract provides good overview of multi-layered approach. The phrasing “delayed differentiation of Sertoli and steroidogenic cell lineages” implies causality; however, causality is shown indirectly and may need softening (Lines 25–29).
Response: Thank you for your suggestion. We have revised the abstract as requested: the loss of DNAH8 function affected differentiation of Sertoli and steroidogenic cell lineages.
Introduction
It is comprehensively literature coverage with strong justification for genetic exploration in hypospadias. The link between androgen deficiency and associated phenotypes is well stated but could benefit from referencing the timing of the masculinization programming window (L# 43-48).
Response: Thank you for your suggestion. We have supplemented this part of the content: When testosterone levels decrease during the human masculine programming window (MPW; 8–14 weeks of gestation), masculinization often appears abnormal.7
The introduction to DNAH8 function relies heavily on infertility studies; more developmental context is needed.
Response: Thank you for your question. As described in the Introduction, current investigations into the function of DNAH8 have primarily relied on infertility research, particularly focusing on its effects on sperm flagella. To date, no studies have explored the role of DNAH8 in development, and thus the present study fills this research gap.
Materials and methods
Methodological breadth is impressive and CRISPR design and genotyping steps are clearly stated. However, following points should be clarified in revision:
Proteomics section needs thresholding details for peptide identification.
Response: Thank you for your suggestion. We have supplemented this part of the content: and the screening criteria for credible proteins and peptides were both set at a False Discovery Rate (FDR) ≤ 0.01.
single-cell processing lacks details on cell viability thresholds and batch correction approach.
Response: Thank you for your question. The details of single-cell processing regarding cell viability thresholds and batch correction approach can be found in the Supplementary Materials and Methods section of the supplementary materials: Dead cells were counted by trypan blue (Sigma Aldrich T8154) on a Bio-Rad TC20 (Bio-Rad, California, USA), and samples with a viability of >85% were subjected to subsequent preparations; Since our fetal rat testes and external genital tissues were obtained and analyzed on the same batch, there is no need for further batch correction using the IntegrateData function in the Seurat package.
Statistical analysis section is too generic; it requires clarity on exact tests for each figure.
Response: Thank you for your suggestion. We have supplemented it in the Figure legends.
Results
The description of urethral plate fusion is solid, but quantification of fusion abnormalities should be shown. The claim that abnormalities disappear by E18.5 may require more animals for statistical confidence.
Response: Thank you for your question. From a morphological and sectional perspective, urethral fusion abnormalities are very objective and clear, and generally do not use quantitative data to display them. Currently, there are three well-recognized research groups worldwide focusing on male urethral malformations, including Gen Yamada, Martin J. Cohn and Humphrey Hung-Chang Yao. These groups have also conducted studies on the genetic and molecular mechanisms underlying hypospadias development. In their research, they also adopted a display of results similar to ours.
- Suzuki K, Numata T, Suzuki H, et al. Sexually dimorphic expression of Mafb regulates masculinization of the embryonic urethral formation. Proc Natl Acad Sci U S A. 2014 Nov 18;111(46):16407-12. doi: 10.1073/pnas.1413273111.
- Amato CM, Xu X, Yao HH. An extragenital cell population contributes to urethra closure during mouse penis development. Sci Adv. 2024 Dec 6;10(49):eadp0673. doi: 10.1126/sciadv.adp0673.
- Kothandapani A, Lewis SR, Noel JL, et al. GLI3 resides at the intersection of hedgehog and androgen action to promote male sex differentiation. PLoS Genet. 2020 Jun 4;16(6):e1008810. doi: 10.1371/journal.pgen.1008810.
We observed a total of 25 E18.5 HOM male fetal mice, and the sample size is sufficient, regarding the claim that abnormalities disappear on day 18.5 of the embryo. In a very classic study in the field of hypospadias research, Martin J. Cohn reported1 that in GLI3 knockout fetal mice, the observed hypospadias phenotype in the early stages was restored with development. We observe a much larger sample size than they do.
- Kothandapani A, Lewis SR, Noel JL, et al. GLI3 resides at the intersection of hedgehog and androgen action to promote male sex differentiation. PLoS Genet. 2020 Jun 4;16(6):e1008810. doi: 10.1371/journal.pgen.1008810.
Proteomic fold-change thresholds should be justified (Lines 247–256).
Response: Thank you for your question. In comparative proteomic studies, a fold change (FC) cutoff of 2.0 is a widely accepted standard. The combination of FC > 2 and P < 0.05 represents a balance between biological significance and statistical reliability, which has been extensively applied in similar studies in developmental biology and reproductive toxicology. For instance, the "FC ≥ 2.0 and P < 0.05" criterion was strictly adopted to screen differentially expressed proteins and metabolites in [Guo Q, Xie M, Wang QN, et al. Comprehensive Serum Proteomic and Metabolomic Profiles of Pediatric Patients with Moyamoya Disease Reveal Core Pathways. J Inflamm Res. 2024 Sep 9;17:6173-6192.].
The explanation for compensation is plausible but speculative; additional functional assays would help (Lines 264–275).
Response: Thank you for your question. We have indeed explored some possible compensatory mechanisms, the most direct of which is the compensatory effect of other ODNAH genes (DNAH5, DNAH9, DNAH11, and DNAH17). DNAH5, DNAH9, and DNAH11—previously identified as causative genes for primary ciliary dyskinesia (PCD)—are predominantly expressed in ciliated epithelial cells.
- Hornef N, Olbrich H, Horvath J, et al. DNAH5 mutations are a common cause of primary ciliary dyskinesia with outer dynein arm defects. Am J Respir Crit Care Med. 2006 Jul 15;174(2):120-6. doi: 10.1164/rccm.200601-084OC.
- Fassad MR, Shoemark A, Legendre M, et al. Mutations in Outer Dynein Arm Heavy Chain DNAH9 Cause Motile Cilia Defects and Situs Inversus. Am J Hum Genet. 2018 Dec 6;103(6):984-994. doi: 10.1016/j.ajhg.2018.10.016.
3.Dougherty GW, Loges NT, Klinkenbusch JA, et al. DNAH11 Localization in the Proximal Region of Respiratory Cilia Defines Distinct Outer Dynein Arm Complexes. Am J Respir Cell Mol Biol. 2016 Aug;55(2):213-24. doi: 10.1165/rcmb.2015-0353OC.
Recent studies have shown that they are also highly expressed in sperm, and their functional loss can lead to male infertility.
- Kamel A, Saberiyan M, Adelian S, et al. DNAH5 gene and its correlation with linc02220 expression and sperm characteristics. Mol Biol Rep. 2022 Oct;49(10):9365-9372. doi: 10.1007/s11033-022-07787-2.
- Zhou H, Yin Z, Ni B, et al. Whole exome sequencing analysis of 167 men with primary infertility. BMC Med Genomics. 2024 Sep 12;17(1):230. doi: 10.1186/s12920-024-02005-
- Guo S, Tang D, Chen Y, et al. Association of novel DNAH11 variants with asthenoteratozoospermia lead to male infertility. Hum Genomics. 2024 Sep 11;18(1):97. doi: 10.1186/s40246-024-00658-w.
DNAH17 has always been considered to be highly expressed in sperm cells, especially in its flagellum. Mutations leading to its functional loss are believed to result in sperm morphology abnormalities and male infertility.
- Zhang B, Ma H, Khan T, et al. A DNAH17 missense variant causes flagella destabilization and asthenozoospermia. J Exp Med. 2020 Feb 3;217(2):e20182365. doi: 10.1084/jem.20182365.
- Whitfield M, Thomas L, Bequignon E, et al. Mutations in DNAH17, Encoding a Sperm-Specific Axonemal Outer Dynein Arm Heavy Chain, Cause Isolated Male Infertility Due to Asthenozoospermia. Am J Hum Genet. 2019 Jul 3;105(1):198-212. doi: 10.1016/j.ajhg.2019.04.015.
In summary, the functions of these four genes in the testes primarily play a role in the postnatal development of sperm and flagella. In our study, we assessed the expression levels of these four genes in fetal mouse testicular tissue. Compared to the control group, no significant compensatory upregulation was observed; in fact, DNAH17 exhibited a decreasing trend, though this difference did not reach statistical significance. Consequently, we conclude that the reversal of the phenotype is unlikely to be due to compensation by the other ODNAH genes (DNAH5, DNAH9, DNAH11, and DNAH17). More likely, it is due to the transient expression of DNAH8 on testicular interstitial progenitors during development. Given that our future research will concentrate on gene–environment interactions, we have not further explored this aspect.
The MM1/MM2 progenitor axis is described well but some lineage intermediates (MM2 → e-IP → FLC) are inferred rather than experimentally validated.
Response: Thank you for your question. The differentiation and developmental pathway (MM2→e-IP→FLC) has been well established in two previous classic studies.1,2 Therefore, we did not perform additional experimental validation and only conducted cell staining observations based on this well-characterized developmental differentiation pathway.
- Ademi H, Djari C, Mayere C, Neirijnck Y, Sararols P et al. Deciphering the origins and fates of steroidogenic lineages in the mouse testis. Cell Rep 2022; 39: 110935.
- Stevant I, Neirijnck Y, Borel C, Escoffier J, Smith LB et al. Deciphering Cell Lineage Specification during Male Sex Determination with Single-Cell RNA Sequencing. Cell Rep 2018; 22: 1589-99.
DEHP Synergy Experiment: Dose justification is reasonable, but pharmacokinetics of DEHP in mice should be noted and sample sizes are relatively small. statistical tests used for incidence comparison should be stated. Also blinded scoring should be mentioned for histology description.
Response: Thank you for your question. Since the damage caused by a single risk gene can be compensated by the body, this study conducted a preliminary exploration of the synergistic effect between genes and the intrauterine environment. Compared with similar studies, our study included more than 20 animals per group, ensuring an adequate sample size. For instance, two classic studies1,2 in the same field used smaller sample sizes per group than ours.
- Wang J, Wei Y, Wu Y, et al. Di-(2-ethylhexyl) phthalate induces prepubertal testicular injury through MAM-related mitochondrial calcium overload in Leydig and Sertoli cell apoptosis. Toxicology. 2024 Dec;509:153956.
- Earl Gray L Jr, Lambright CS, Evans N, et al. Using targeted fetal rat testis genomic and endocrine alterations to predict the effects of a phthalate mixture on the male reproductive tract. Curr Res Toxicol. 2024 Jun 11;7:100180.
We are currently conducting further research to investigate the synergistic effect of prenatal DEHP exposure and DNAH8 loss-of-function in the pathogenesis of hypospadias. As suggested, we will incorporate the pharmacokinetic profiles of DEHP in mice into this follow-up study. We anticipate presenting our future research findings in due course.
Regarding the need to specify the statistical test method used for comparing incidence rates, we fully agree with your suggestion and have supplemented this information in the Results section Figure 7: (A) The incidence of hypospadias from WT (n=35), WT+DEHP250mg/kg (n=20), HOM (n=20) and HOM+DEHP250mg/kg (n=29) group of E16.5 mice (Chi-square test). *P < 0.05, ns: no significance.
The purpose of blinded scoring is to avoid observer bias and ensure the objectivity and reliability of experimental results. As shown in Figure 7C, the phenotypes of urethral fusion (normal vs. abnormal) observed in this study are highly distinguishable and objective, thus blinded scoring was not deemed necessary.
Discussion
This section Logically integrates findings. Following points should be addressed in revised version:
Clarify why phenotypes appear only at E16.5 but not E18.5 (Lines 426–431).
Response: Thank you for your question and this is also a pain point in the study of hypospadias. In early studies on hypospadias, researchers hoped to identify key pathogenic single genes responsible for the condition. A 2012 review, which included 169 studies, analyzed genetic screenings for single-gene defects in patients with hypospadias. The findings revealed that mutations in WT1, SF1, BMP4, BMP7, HOXA4, HOXB6, FGF8, FGFR2, AR, HSD3B2, SRD5A2, ATF3, MAMLD1, MID1, and BNC2 were associated with the development of hypospadias.
van der Zanden LF, van Rooij IA, Feitz WF, et al. Aetiology of hypospadias: a systematic review of genes and environment. Hum Reprod Update. 2012 May-Jun;18(3):260-83. doi: 10.1093/humupd/dms002.
However, many of these gene mutations were not replicated in subsequent animal studies. Alternatively, knockout (KO) mice exhibited severe urethral defects during early embryonic stages prior to sexual differentiation, often resulting in embryonic lethality.
- Blaschko SD, Cunha GR, Baskin LS. Molecular mechanisms of external genitalia development. Differentiation. 2012 Oct;84(3):261-8. doi: 10.1016/j.diff.2012.06.003.
- Petiot A, Perriton CL, Dickson C, et al. Development of the mammalian urethra is controlled by Fgfr2-IIIb. Development. 2005 May;132(10):2441-50. doi: 10.1242/dev.01778.
- Kalfa N, Sultan C, Baskin LS. Hypospadias: etiology and current research. Urol Clin North Am. 2010 May;37(2):159-66. doi: 10.1016/j.ucl.2010.03.010.
- Cunha GR, Sinclair A, Risbridger G, et al. Current understanding of hypospadias: relevance of animal models. Nat Rev Urol. 2015 May;12(5):271-80. doi: 10.1038/nrurol.2015.57.
Currently, there are three well-recognized research groups worldwide focusing on male urethral malformations, including Gen Yamada, Martin J. Cohn and Humphrey Hung-Chang Yao. These groups have also conducted studies on the genetic and molecular mechanisms underlying hypospadias development.
Yamada1 identified the v-maf avian musculoaponeurotic fibrosarcoma oncogene homolog B (Mafb) gene that is significantly expressed in the stroma of male genital nodules (GT),and revealed that Mafb KO male GTs exhibit defective embryonic urethral formation, giving insight into the common human congenital anomaly hypospadias. Yao2 discovered a group of "extragenital" cells marked by the lineage marker Nr5a1, which migrated from the inguinal region into the embryonic penis and facilitate urethra closure by interacting with adjacent periurethral cells via the epidermal growth factor pathway. Ablation of Nr5a1+ cells leads to severe hypospadias, shedding light on the biology of penis formation and potential implications for human hypospadias. Cohn's research3 showed a novel function for the activated form of GLI3 that translates Hedgehog signals to reinforce fetal Leydig cell identity and stimulate timely INSL3 and testosterone synthesis in the developing testis. And they found that Gli3XtJ mutant mice exhibit cryptorchidism and hypospadias due to local effects of GLI3 loss and systemic effects of testicular hormone deficiency.
- Suzuki K, Numata T, Suzuki H, et al. Sexually dimorphic expression of Mafb regulates masculinization of the embryonic urethral formation. Proc Natl Acad Sci U S A. 2014 Nov 18;111(46):16407-12. doi: 10.1073/pnas.1413273111.
- Amato CM, Xu X, Yao HH. An extragenital cell population contributes to urethra closure during mouse penis development. Sci Adv. 2024 Dec 6;10(49):eadp0673. doi: 10.1126/sciadv.adp0673.
- Kothandapani A, Lewis SR, Noel JL, et al. GLI3 resides at the intersection of hedgehog and androgen action to promote male sex differentiation. PLoS Genet. 2020 Jun 4;16(6):e1008810. doi: 10.1371/journal.pgen.1008810.
Although their work is highly commendable, certain limitations remain. The genes or cellular abnormalities they identified were not derived from screenings of clinical patients, and their occurrence in affected individuals is relatively low. Meanwhile, mutations in some genes, in addition to hypospadias, can also lead to a syndrome of multiple organ malformations. Therefore, this insufficiency fails to account for the high prevalence of hypospadias, particularly the increasing incidence of isolated hypospadias cases.
In recent years, advancements in exome and whole-genome sequencing technologies have created new opportunities, particularly for studying the growing number of patients with isolated hypospadias. Concurrently, as our understanding of the genetic landscape underlying hypospadias continues to progress, it is now widely recognized that hypospadias is a multifactorial disorder. The primary challenge in uncovering the mechanisms of hypospadias lies in focusing on gene-environment interactions.
van der Zanden LF, van Rooij IA, Feitz WF, et al. Aetiology of hypospadias: a systematic review of genes and environment. Hum Reprod Update. 2012 May-Jun;18(3):260-83. doi: 10.1093/humupd/dms002.
As you mentioned, although the expected hypospadias phenotype was not consistently observed in the postnatal stage of DNAH8 gene knockout (KO) mice in our study, and this inconsistency raises questions about the robustness of the model and the degree to which the DNAH8 gene alone can be attributed to hypospadias pathogenesis. However, DNAH8, a newly identified risk gene for hypospadias, was discovered through whole-exome sequencing (WES) of clinical patients. The results from the mouse model further validate the concept that hypospadias, particularly isolated hypospadias, is not caused by a single gene but results from a combination of multiple genetic and environmental factors. Recent articles on the role of Endocrine disrupting chemicals (EDCs), in urethral development and fusion also indicate that the intrauterine environment plays an important role in the occurrence of hypospadias. And except for serious defects such as Mafb and Nr5a1, most genes or environmental factors1-4 only produce a certain proportion of hypospadias phenotype in animal models (27.6% - 81%).
- Shi B, He E, Chang K, et al. Genistein prevents the production of hypospadias induced by Di-(2-ethylhexyl) phthalate through androgen signaling and antioxidant response in rats. J Hazard Mater. 2024 Mar 15;466:133537. doi: 10.1016/j.jhazmat.2024.133537.
- Kothandapani A, Lewis SR, Noel JL, et al. GLI3 resides at the intersection of hedgehog and androgen action to promote male sex differentiation. PLoS Genet. 2020 Jun 4;16(6):e1008810. doi: 10.1371/journal.pgen.1008810.
- Zhou Y, Huang F, Liu Y, et al. TGF-β1 relieves epithelial-mesenchymal transition reduction in hypospadias induced by DEHP in rats. Pediatr Res. 2020 Mar;87(4):639-646. doi: 10.1038/s41390-019-0622-2.
- Cripps SM, Mattiske DM, Black JR, et al. A loss of estrogen signaling in the aromatase deficient mouse penis results in mild hypospadias. Differentiation. 2019 Sep-Oct;109:42-52. doi: 10.1016/j.diff.2019.09.001.
In DNAH8 knockout mice, hypospadias was observed at E16.5 but was not consistently present at E18.5. We propose that this finding may better reflect real-world conditions, where the occurrence of hypospadias is more likely influenced by gene-environment interactions. Therefore, we have provided an in-depth analysis of the potential interplay between genetic factors and intrauterine exposure such as EDCs in contributing to the pathogenesis of hypospadias. We have elaborated on this part in the highlighted red font paragraph of the discussion section.
The argument about transient motile cilia requires citations showing motile cilia presence in early testis progenitors (Lines 514–529).
Response: Thank you for your question. To date, research on the role of motile cilia during development has primarily focused on the nervous and respiratory systems, with a research gap remaining in testicular development—particularly in early testis progenitors. DNAH8 is well-established to be expressed on motile cilia, and we further identified its expression in early testis progenitors (within the MM and IP cell populations), albeit at a relatively low level (Figure S9A).
The gene × environment model discussion is excellent but could be condensed (Lines 538–562).
Response: Thank you for your suggestion. We have condensed the last paragraph of the discussion section.
Reviewer 3 Report
Comments and Suggestions for Authors
The manuscript entitled "Prenatal low testosterone levels induced by DNAH8 dysfunction leads to urethral fusion and male differentiation abnormalities" is well written in clear understandable English. Authors focuses the attention of the reader on the problem of hypospadia, the investigate knockout mice exhibiting hypospadia signs during intrauterine development. Authors performed interesting work and in general it is well-performed paper, however I have some questions and recommendations:
1. Chapter 2.1. Were the mice created by injection of crispr/cas9 in PN of zygotes or by ESCs technology?
2. DNAH8 homozygous males were infertile due to the inability to perform the coitus or due to the absence of spermatozoa? Did you observed vaginal plugs? How many female mice were breed per one male?
3. Embryos at E16.5 can suffer form pain. How did you anesthesized the embryos? Provide protocol of Bioethical committee that your work was approved.
4. Too many information is in Supplemetary materials. It will be better to introduce it in the main text of the manuscript: antibodies, primers, protocols.
5. Line 125 - HOM - homozygote?
6. Were some abnormalities in females?
7. Figure S1 - where is scale bar?
8. Figure S8. In the presented image there is no mature spermatozoa in the seminiferous. Did you performed spermogramm for such mice? Or did you analyzed testis for the presence of normal spermatozoa by histological sectioning?
9. Line 157. It is the start of the phrase. Use W.
10. Make the separate table with primers uised. Indicate NM-numbers, product length, melting temperature.
11. Line 275 "While in the late stage of E18.5, this abnormal decrease was reversed because of the compensatory mechanism. " Please, propose how this mechanism may function? which molecular pathways may be involved?
12. It will be interesting to see experiments with some chemicals which allow to rescue hypospadia phenotype in homozygous mice.
Author Response
- Chapter 2.1. Were the mice created by injection of crispr/cas9 in PN of zygotes or by ESCs technology?
Response: Thank you for your question. The mice were created by injection of CRISPR/Cas9 in PN of zygotes: Homozygote whole-body DNAH8 knockout mice (C57BL/6J) were established by Shanghai Southern Model Biotechnology, China. CRISPR/Cas9 technology was employed, utilizing non-homologous recombination repair to introduce mutations, leading to a frameshift in the DNAH8 protein coding sequence and ultimately resulting in a loss of function. One sgRNA targeting intron 3 of the DNAH8 gene was 5'-AGACTTAAGACCTCCGAGG-3', and the other sgRNA targeting intron 4 of the DNAH8 gene was 5'-CATTACTTAUGUACACTCTGGG-3'.
DNAH8 homozygous males were infertile due to the inability to perform the coitus or due to the absence of spermatozoa? Did you observed vaginal plugs? How many female mice were breed per one male?
Response: Thank you for your question. Asthenozoospermia is a common cause of male infertility associated with the reduced motility and/or abnormal morphology of spermatozoa. MMAF is one of the main causes of asthenozoospermia. Previous studies have extensively explored the role of DNAH8 and other DNAH family genes in the pathogenesis of MMAF and infertility, which has been widely acknowledged. The specific references are as follows:
- Liu C, Miyata H, Gao Y, et al. Bi-allelic DNAH8 Variants Lead to Multiple Morphological Abnormalities of the Sperm Flagella and Primary Male Infertility. Am J Hum Genet. 2020 Aug 6;107(2):330-341. doi: 10.1016/j.ajhg.2020.06.004.
- Yang Y, Jiang C, Zhang X, et al. Loss-of-function mutation in DNAH8 induces asthenoteratospermia associated with multiple morphological abnormalities of the sperm flagella. Clin Genet. 2020 Oct;98(4):396-401. doi: 10.1111/cge.13815.
- Tang D, Sha Y, Gao Y, et al. Novel variants in DNAH9 lead to nonsyndromic severe asthenozoospermia. Reprod Biol Endocrinol. 2021 Feb 20;19(1):27. doi: 10.1186/s12958-021-00709-0.
- Zhang B, Ma H, Khan T, et al. A DNAH17 missense variant causes flagella destabilization and asthenozoospermia. J Exp Med. 2020 Feb 3;217(2):e20182365. doi: 10.1084/jem.20182365.
Overall, sperm flagella are classified as motile cilia, with a characteristic 9+2 ultrastructure consisting of two central parallel microtubules surrounded by nine double stranded microtubules. These peripheral double stranded microtubules are connected by linking proteins and dynein arms, presenting a wheel like structure, which are essential for flagella motility. Loss of function in DNAH8 or other DNAH family genes leads to flagellar abnormalities, resulting in impaired sperm motility and, ultimately, infertility. Furthermore, existing studies strongly suggest that intracytoplasmic sperm injection (ICSI) is a promising therapeutic approach. The role of DNAH8 and other DNAH family genes in MMAF-associated phenotypes and their contributions to male infertility have been extensively investigated. Consistent with these findings, our study confirmed that HOM male mice are infertile, making additional redundant experiments unnecessary.
Our team's research primarily focuses on congenital urogenital anomalies. DNAH8 was identified through whole-exome sequencing as a novel risk gene for hypospadias. We aim to explore its role in urethral development. To date, no studies from other research groups have reported the involvement of DNAH8 or other DNAH family genes in hypospadias, making this an innovative aspect of our research. This study provides new insights into the role of cilia-related genes in urethral development, thereby expanding the understanding of their contribution beyond known functions.
:We observed vaginal plugs and 1:1 reproduction between male and female mice: Each female mouse was mated with another male adult mouse at 5 pm and the morning when the vaginal plug was found was considered embryonic day 0 (E0).
Embryos at E16.5 can suffer form pain. How did you anesthesized the embryos? Provide protocol of Bioethical committee that your work was approved.
Response: Thank you for your question. We did not administer anesthesia to the embryos. As in previous similar studies,1,2 when harvesting E16.5 fetal mice, they were euthanized by routine cervical dislocation followed by immediate dissection for tissue collection.
- Kothandapani A, Lewis SR, Noel JL, Zacharski A, Krellwitz K et al. GLI3 resides at the intersection of hedgehog and androgen action to promote male sex differentiation. PLoS Genet 2020; 16: e1008810.
- Ademi H, Djari C, Mayere C, Neirijnck Y, Sararols P et al. Deciphering the origins and fates of steroidogenic lineages in the mouse testis. Cell Rep 2022; 39: 110935.
Too many information is in Supplemetary materials. It will be better to introduce it in the main text of the manuscript: antibodies, primers, protocols.
Response: Thank you for your question. Due to the word count limit required by the journal, we have included details such as antibodies, primers, and experimental protocols in the supplementary materials. We have streamlined the supplementary materials in accordance with your comments.
Line 125 - HOM - homozygote?
Response: Thank you for your question. Yes, HOM is homozygous. And we have revised this sentence: For each WT and homozygous (HOM) sample, three technical replicates were analyzed.
- Were some abnormalities in females?
Response: Thank you for your question. We also observed female mice and found no abnormalities in their development and reproduction.
Figure S1 - where is scale bar?
Response: Thank you for your valuable suggestion. We have revised Figure S1 and added a scale bar.
Figure S8. In the presented image there is no mature spermatozoa in the seminiferous. Did you performed spermogramm for such mice? Or did you analyzed testis for the presence of normal spermatozoa by histological sectioning?
Response: Thank you for your question. The DNAH8 we stained is expressed on the flagella of spermatozoa. Notably, compared with wild-type (WT) and heterozygous (HET) mice, no stained sperm flagella were observed in the seminiferous tubules of homozygous (HOM) male mice. As answered in question 2, asthenozoospermia is a common cause of male infertility associated with the reduced motility and/or abnormal morphology of spermatozoa. MMAF is one of the main causes of asthenozoospermia. Previous studies have extensively explored the role of DNAH8 and other DNAH family genes in the pathogenesis of MMAF and infertility, which has been widely acknowledged. The specific references are as follows:
- Liu C, Miyata H, Gao Y, et al. Bi-allelic DNAH8 Variants Lead to Multiple Morphological Abnormalities of the Sperm Flagella and Primary Male Infertility. Am J Hum Genet. 2020 Aug 6;107(2):330-341. doi: 10.1016/j.ajhg.2020.06.004.
- Yang Y, Jiang C, Zhang X, et al. Loss-of-function mutation in DNAH8 induces asthenoteratospermia associated with multiple morphological abnormalities of the sperm flagella. Clin Genet. 2020 Oct;98(4):396-401. doi: 10.1111/cge.13815.
- Tang D, Sha Y, Gao Y, et al. Novel variants in DNAH9 lead to nonsyndromic severe asthenozoospermia. Reprod Biol Endocrinol. 2021 Feb 20;19(1):27. doi: 10.1186/s12958-021-00709-0.
- Zhang B, Ma H, Khan T, et al. A DNAH17 missense variant causes flagella destabilization and asthenozoospermia. J Exp Med. 2020 Feb 3;217(2):e20182365. doi: 10.1084/jem.20182365.
Overall, sperm flagella are classified as motile cilia, with a characteristic 9+2 ultrastructure consisting of two central parallel microtubules surrounded by nine double stranded microtubules. These peripheral double stranded microtubules are connected by linking proteins and dynein arms, presenting a wheel like structure, which are essential for flagella motility. Loss of function in DNAH8 or other DNAH family genes leads to flagellar abnormalities, resulting in impaired sperm motility and, ultimately, infertility. Furthermore, existing studies strongly suggest that intracytoplasmic sperm injection (ICSI) is a promising therapeutic approach. The role of DNAH8 and other DNAH family genes in MMAF-associated phenotypes and their contributions to male infertility have been extensively investigated. Consistent with these findings, our study confirmed that HOM male mice are infertile and mentioned in the Results section, making additional redundant experiments unnecessary: Male mice in the HOM group were mated for more than 2 months and failed to give birth to offspring, confirming that biallelic DNAH8 mutation causes sterility.17
Line 157. It is the start of the phrase. Use W.
Response: Thank you for your suggestion and we have corrected this mistake.
Make the separate table with primers used. Indicate NM-numbers, product length, melting temperature.
Response: Thank you for your valuable suggestion to organize the primers used in this study into a separate table with detailed information. We have complied with your request and prepared a comprehensive primer table (Table S1, Supplementary Materials).
Line 275 "While in the late stage of E18.5, this abnormal decrease was reversed because of the compensatory mechanism. " Please, propose how this mechanism may function? which molecular pathways may be involved?
Response: Thank you for your question. We have indeed explored some possible compensatory mechanisms, the most direct of which is the compensatory effect of other ODNAH genes (DNAH5, DNAH9, DNAH11, and DNAH17). DNAH5, DNAH9, and DNAH11—previously identified as causative genes for primary ciliary dyskinesia (PCD)—are predominantly expressed in ciliated epithelial cells.
- Hornef N, Olbrich H, Horvath J, et al. DNAH5 mutations are a common cause of primary ciliary dyskinesia with outer dynein arm defects. Am J Respir Crit Care Med. 2006 Jul 15;174(2):120-6. doi: 10.1164/rccm.200601-084OC.
- Fassad MR, Shoemark A, Legendre M, et al. Mutations in Outer Dynein Arm Heavy Chain DNAH9 Cause Motile Cilia Defects and Situs Inversus. Am J Hum Genet. 2018 Dec 6;103(6):984-994. doi: 10.1016/j.ajhg.2018.10.016.
3.Dougherty GW, Loges NT, Klinkenbusch JA, et al. DNAH11 Localization in the Proximal Region of Respiratory Cilia Defines Distinct Outer Dynein Arm Complexes. Am J Respir Cell Mol Biol. 2016 Aug;55(2):213-24. doi: 10.1165/rcmb.2015-0353OC.
Recent studies have shown that they are also highly expressed in sperm, and their functional loss can lead to male infertility.
- Kamel A, Saberiyan M, Adelian S, et al. DNAH5 gene and its correlation with linc02220 expression and sperm characteristics. Mol Biol Rep. 2022 Oct;49(10):9365-9372. doi: 10.1007/s11033-022-07787-2.
- Zhou H, Yin Z, Ni B, et al. Whole exome sequencing analysis of 167 men with primary infertility. BMC Med Genomics. 2024 Sep 12;17(1):230. doi: 10.1186/s12920-024-02005-
- Guo S, Tang D, Chen Y, et al. Association of novel DNAH11 variants with asthenoteratozoospermia lead to male infertility. Hum Genomics. 2024 Sep 11;18(1):97. doi: 10.1186/s40246-024-00658-w.
DNAH17 has always been considered to be highly expressed in sperm cells, especially in its flagellum. Mutations leading to its functional loss are believed to result in sperm morphology abnormalities and male infertility.
- Zhang B, Ma H, Khan T, et al. A DNAH17 missense variant causes flagella destabilization and asthenozoospermia. J Exp Med. 2020 Feb 3;217(2):e20182365. doi: 10.1084/jem.20182365.
- Whitfield M, Thomas L, Bequignon E, et al. Mutations in DNAH17, Encoding a Sperm-Specific Axonemal Outer Dynein Arm Heavy Chain, Cause Isolated Male Infertility Due to Asthenozoospermia. Am J Hum Genet. 2019 Jul 3;105(1):198-212. doi: 10.1016/j.ajhg.2019.04.015.
In summary, the functions of these four genes in the testes primarily play a role in the postnatal development of sperm and flagella. In our study, we assessed the expression levels of these four genes in fetal mouse testicular tissue. Compared to the control group, no significant compensatory upregulation was observed; in fact, DNAH17 exhibited a decreasing trend, though this difference did not reach statistical significance. Consequently, we conclude that the reversal of the phenotype is unlikely to be due to compensation by the other ODNAH genes (DNAH5, DNAH9, DNAH11, and DNAH17). More likely, it is due to the transient expression of DNAH8 on testicular interstitial progenitors during development. Given that our future research will concentrate on gene–environment interactions, we have not further explored this aspect.
It will be interesting to see experiments with some chemicals which allow to rescue hypospadias phenotype in homozygous mice.
Response: Thank you for your suggestion. We have acknowledged this limitation and are currently focusing on exploring the synergistic effects of prenatal DEHP exposure and DNAH8 loss-of-function in the pathogenic mechanism of hypospadias in our follow-up studies. Fortunately, we have obtained some preliminary results. Further, in line with your expert advice, we will design experiments to investigate whether the hypospadias phenotype in homozygous mice can be rescued. We anticipate presenting our future research findings in due course.
Round 2
Reviewer 3 Report
Comments and Suggestions for Authors
Can be accepted.